# Generation of Polyamide 12 Coatings on Stainless Steel Substrates by Directed Energy Deposition with a Thulium-Doped Fiber Laser (DED-LB/P)

**DOI:** 10.3390/polym14183729

**Published:** 2022-09-07

**Authors:** Alexander Wittmann, Oliver Hentschel, Alexander Sommereyns, Michael Schmidt

**Affiliations:** 1Institute of Photonic Technologies (LPT), Friedrich-Alexander-Universität Erlangen-Nürnberg, Konrad-Zuse-Straße 3/5, 91052 Erlangen, Germany; 2Erlangen Graduate School in Advanced Optical Technologies (SAOT), Friedrich-Alexander-Universität Erlangen-Nürnberg, Paul-Gordan-Straße 6, 91052 Erlangen, Germany; 3Bayerisches Laserzentrum GmbH (BLZ), Konrad-Zuse-Straße 2/6, 91052 Erlangen, Germany

**Keywords:** DED, directed energy deposition, laser polymer deposition, PA12, thulium fiber laser, polymer coating

## Abstract

Due to their good material properties (e.g., corrosion and wear resistance, biocompatibility), thermoplastic materials like polyamide 12 (PA12) are interesting for functional coatings on metallic components. To ensure a spatially resolved coating and to shorten the process chain, directed energy deposition of polymer powders by means of a laser beam (DED-LB/P) offers a promising approach. Due to characteristic absorption bands, the use of a thulium fiber laser with a wavelength of 1.94 μm is investigated in a DED-LB/P setup to generate PA12 coatings on stainless steel substrates without the need to add any absorbing additives. The influence of the energy density and powder mass flow was analyzed by infrared thermography. Furthermore, the coatings were characterized by differential scanning calorimetry, laser-scanning-microscopy, optical microscopy and cross-cutting tests. The results in this study demonstrate for the first time the basic feasibility of an absorber-free DED-LB/P process by using a thulium fiber laser. PA12 coatings with a low porosity and good adhesion are achievable. Depending on the application-specific requirements, a trade-off must be made between the density and surface quality of the PA12 coatings. The use of infrared thermography is appropriate for in-situ detection of process instabilities caused by an excessive energy input.

## 1. Introduction

Functional coatings made of thermoplastics such as polyamide 12 (PA12) or polyether ether ketone (PEEK) are known for their biocompatibility, excellent chemical resistance, low sliding friction coefficients and high abrasion resistance [1]. Thus, thermoplastic coatings are particularly suitable to avoid corrosion and excessive wear of metallic surfaces in demanding environments [2].

Thermoplastics are commonly deposited onto metallic components by electrostatic powder spraying or dispersion coating (spray gun, dipping). Afterward, the coated part is heated above the melting temperature of the polymer in a furnace [3]. This thermal post-treatment is not mandatory for coatings applied by the flame spraying technique [4]. However, a high porosity and weak adhesion strength between the coatings and the base material can be observed [5].

In order to reduce the heat input into the base material and ensure a spatially resolved polymer deposition, different laser-based coating processes are the subject of scientific studies [3,5,6,7]. A limiting factor with respect to the process efficiency is, analogously to the conventional coating processes, that the powder deposition and laser-based consolidation are carried out separately. Moreover, the powder deposition by a recoater mechanism is not suited to coat parts with complex shaped geometries and has therefore a low application potential in industry.

For addressing these challenges, directed energy deposition (DED) represents a promising approach, where the feeding and fusing of the material (powder or wire) occur quasi-simultaneously using a heat source such as a laser or plasma arc [8]. DED of metallic powders by means of a laser beam (DED-LB/M) is established for generating wear or corrosion resistant coatings. Furthermore, the repair of worn out parts (e.g., turbine blades, forming tools) and manufacturing of near-net-shape components are industrial applications for DED-LB/M [9]. The use of a multi-hopper powder feeder in a DED system allows the spatial variation of the chemical composition to produce functionally graded components. In comparison to laser powder bed fusion (PBF-LB), the part size is generally not limited by the build volume, and the deposition rates are higher [8].

Compared to DED-LB/M, only a few studies about DED of polymer powders by means of a laser beam (DED-LB/P) are published in the literature. Polyamide 11 (PA11) and thermoplastic polyurethane (TPU) based coatings were applied by DED-LB/P on TPU and nitrile butadiene rubber substrates [10,11]. For this purpose, a diode laser with a wavelength of 940 nm was used to consolidate the polymer coatings. Polytetrafluoroethylene (PTFE) and molybdenum disulfide (MoS_2_) powders, which are well-known solid lubricants, were mixed with the feedstocks for reducing the dry sliding friction. The incorporation of 15 wt.% PTFE in the PA11 coating resulted in the largest decrease of the frictional forces [11]. In contrast, TPU based coatings applied by DED-LB/P did not reduce friction forces [10]. Moreover, the basic feasibility of manufacturing three-dimensional (3D) structures by DED-LB/P is proven by previous authors. A DED-LB/M setup consisting of a coaxial powder nozzle and a near-infrared (NIR) solid-state laser was utilized for processing PA12 [12,13] and TPU powders [14]. The rotating plate powder feeder was modified with an additional vibrating module to avoid powder agglomerations and hence ensure a reproducible powder mass flow [12]. Results show weak mechanical properties [13] and a poor surface quality of the 3D structures [14], which are caused by partially molten particles and high porosity. Therefore, improvements on the material as well as the process side are necessary to achieve a competitive additive manufacturing technology.

In general, polymers exhibit a high transparency for the radiation emitted by NIR laser beam sources in the wavelength range between 0.8 µm and 1.1 µm [15]. For this reason, the thermoplastic feedstocks contained additives (inter alia carbon black [10,14] or multi-walled carbon nanotubes [12,13]) in the previous investigations about DED-LB/P to enhance the laser absorption. However, the use of these additives requires an additional processing step for the application and therefore raises the total manufacturing costs [16]. Furthermore, the biocompatibility of the printed parts can be affected, which inhibits the use in particular industries such as the medical sector [17].

Besides the adaptation of the polymer system, the variation of the laser beam source for the DED-LB/P process is a further possibility to ensure suitable absorption characteristics. In laser powder bed fusion of polymers (PBF-LB/P), CO_2_ lasers are mainly used due to the high absorption of polymers at the wavelength of 10.6 µm [18]. However, it must be considered from a technological perspective that the laser radiation at 10.6 µm cannot be guided through glass fibers, which makes the integration into a DED-LB/P system more challenging than with NIR laser beam sources [14].

As most thermoplastic polymers show characteristic absorption bands at wavelengths higher than 1.5 µm [19], the integration of a thulium-doped fiber laser system with a wavelength of around 2 µm in a DED-LB/P setup is a promising approach. This fiber-coupled laser beam source is commonly used for absorber-free laser transmission welding of polymers and biopolymers [16]. Laumer et al. [20] examined the optical properties of thermoplastic powders by a measurement setup consisting of two integration spheres and a laser source with a wavelength of 1.94 µm. PA12 powder layers with a thickness of 200 µm showed an absorptance of 31%, a reflectance of 40% and a transmittance of 29%. In our previous study [7], a two-step coating process was investigated, which includes the manual deposition of PA12 powder by a recoating mechanism and a laser-based consolidation by means of a thulium-doped fiber laser with a wavelength of 1.94 µm. The suitability of a thulium-doped fiber laser was proven by producing dense PA12 coatings with an adhesion to stainless steel substrates.

In the present study, the use of a thulium-doped fiber laser with a wavelength of 1.94 μm is investigated in a DED-LB/P setup for the first time to consolidate PA12 coatings on stainless steel substrates. Based on the aforementioned optical characteristics of the polymer powder, it is assumed that a thulium-doped fiber laser is appropriate for fabricating dense and adherent PA12 coatings on stainless steel substrates by DED-LB/P. We aim to understand the influence of the laser power and powder mass flow on the processability and resulting coating characteristics (layer thickness, crystallinity, surface topography, porosity and coating adhesion). As a result, the feasibility of an absorber-free DED-LB/P process by using a thulium-doped fiber laser will be proven, and process strategies for defect-free coatings will be derived.

## 2. Materials and Methods

### 2.1. Powder and Substrate Material

Commercially available PA12 powder (PA 2200, EOS, Krailling, Germany) with an average particle size ($x_{50}$) of 58 μm, according to the manufacturer’s data sheet, was used for the DED-LB/P process due to its good powder flowability and large processing window. Based on the high relevance of stainless steels in various industrial and biomedical applications, AISI 316L was selected exemplarily as substrate material to show the feasibility of DED-LB/P on stainless steels. Substrates with a thickness of 2 mm and a quadratic shape (50 × 50 mm^2^) were utilized for the experimental investigations. In order to increase the roughness of the substrates and hence promote the mechanical interlocking between the substrate and coating [21], the substrate surfaces were sandblasted (SM 2002 A, HGH, Lüdenscheid, Germany) using a microblasting system (HGH 6040, HGH, Lüdenscheid, Germany). The average roughness (Sa) of the sandblasted surfaces was determined as 2.10 ± 0.15 μm. Residues of sandblasting on the substrates were removed by compressed air.

### 2.2. DED-LB/P Setup

In this study, the DED-LB/P process was performed with the experimental setup schematically shown in Figure 1. The beam source was a single-mode thulium-doped fiber laser system (TLR-120, IPG, Burbach, Germany) emitting laser light with a wavelength of 1.94 μm and a maximum continuous-wave (cw) output power of 120 W. The laser processing head (YC30, Precitec, Gaggenau, Germany) consists of a fiber adapter, focusing lens, protective window and four-jet nozzle. The laser beam (Gaussian profile) was guided onto the substrate with a spot diameter of 1.9 mm. The powder feed to the four-jet nozzle of the laser processing head was accomplished using a vibrating powder feeder (Flowmotion, Medicoat, Wohlen, Switzerland). Argon was selected as a carrier as well as shielding gas to avoid oxidative degradation mechanisms of the polymer. The substrate plate was positioned by an *x*–*y*-motion stage (Aerotech, Fürth, Germany) to deposit the polymer powder in a meander-shaped pattern. The working distance between the substrate and the nozzle tip was adjusted by moving the laser processing head (*z*-motion), which is mounted on a linear *z*-axis (Aerotech, Fürth, Germany).

Preliminary feeding experiments show a reproducible adjustable powder mass flow of the PA12 powder, which can be associated with a small relative standard deviation (≤3.2%) of the measured powder mass values, see Figure 2a. This reproducibility is an important prerequisite for a stable DED-LB/P process. For evaluating the powder stream characteristics, high-speed camera videos are analyzed as demonstrated in Figure 2b. Values of 2.8 mm and 1.2 mm for the full width at half maximum in vertical (${FWHM}_{V}$) and horizontal directions (${FWHM}_{H}$) were calculated, respectively. The distance (l) between the nozzle tip and powder stream focus was determined as 13.6 mm. This value corresponds to the working distance being equivalent to the distance between the nozzle tip and substrate surface in DED-LB/P processing.

### 2.3. Deposition of PA12 Coatings

Table 1 presents the investigated process parameters for generating PA12 coatings by DED-LB/P. The laser power and powder mass flow were varied to investigate their influence on the process as well as the coating properties (layer thickness, crystallinity, surface topography, porosity and coating adhesion). The hatch distance, feed rate, beam diameter, carrier gas flow and shielding gas flow were kept constant. The corresponding energy density $E_{D}$ was calculated by Equation (1), including the experimental process parameters of laser power P, feed rate v and beam diameter d [22]:(1)$$E_{D\;}=\frac{P}{v\;\cdot \;d}$$

In order to monitor the thermal radiation emitted during DED-LB/P processing, an infrared (IR) camera (VarioCAM HD head 600, Infratec, Dresden, Germany) was used, which was mounted on the *z*-axis of the experimental DED-LB/P setup (Figure 1). The utilized camera with an uncooled microbolometer focal plane array is capable of detecting IR radiation in a wavelength range of 7.5 µm to 14 µm and is calibrated in the temperature range between −40 °C and 2000 °C. The camera was operated at a frequency of 60 Hz with a spatial resolution of 640 × 480 pixels. The angle between the optical axis of the camera and the surface of the substrate was approximately 45°. The distance between the front lens of the IR camera and the laser spot on the substrate was around 250 mm, which results in a spatial resolution of about 0.21 mm pixel edge length. A protective germanium window, which is impermeable for the wavelength of the thulium fiber laser, is employed in front of the camera to prevent damaging the detector. In accordance with literature [23], the emission coefficient was set to a value of 0.98 for all experiments. Due to the temperature dependency of the emissivity, the measured temperatures are not considered as absolute values. However, the influence of the process parameters on the DED-LB/P process and resulting characteristics can be identified.

### 2.4. Analysis of the PA12 Powder and Coatings

Non-isothermal differential scanning calorimetry (DSC) tests were conducted using a DSC822e machine (Mettler Toledo, Columbus, OH, USA) under a nitrogen purge of 40 mL/min for evaluating thermal features of both PA12 powder as well as coatings. The solidified PA12 layers were removed from the substrate by means of a scalpel and samples with a mass of about 6 mg were placed in 40 μL aluminum pans with covers. To ensure equal thermal conditions at the start of the measurement, the samples were heated from 25 °C to 80 °C at a heating rate of 20 K/min and held for 3 min at 80 °C. Subsequently, the samples were heated from 80 °C to 230 °C at a rate of 10 K/min with a holding time of 3 min at 230 °C to eliminate residual crystals. Finally, the melt was cooled down to 25 °C at a cooling rate of 10 K/min. An integral horizontal baseline was used to evaluate the enthalpy of fusion $\Delta H_{m}$. The crystallinity $X_{C\;}$ of the samples was calculated according to Equation (2) [24]. The enthalpy of fusion of a theoretical 100% crystalline PA12 material $\Delta H_{100}$ of 209.3 J/g was taken from literature [25]:(2)$$X_{C\;}=\frac{\Delta H_{m}}{\Delta H_{100}}$$

The layer height of the fabricated coating layers was determined for each parameter combination by a dial gauge (ID-S1012B, Mitutoyo, Kawasaki, Japan). The surface topography of the PA12 coatings was characterized by a confocal laser scanning microscope (LSM) LEXT 4000 (Olympus, Tokyo, Japan). For this work, the LSM images of the surfaces are taken with a magnification of 10×. The open-source software Gwyddion was used to analyze and visualize the data from LSM.

In a further step, cross-sections were prepared using a disc cutting machine (Discotom-10, Struers, Willich, Germany) and subsequently grinded as well as polished. The generated cross-sections were characterized in terms of possible defects by optical microscopy (BX53M, Olympus, Tokyo, Japan). With the help of the open-source image processing software ImageJ, the porosity of the coating layers was determined.

For assessing the adhesion of the coating on the substrate, cross-cutting tests according to DIN EN ISO 2409 were conducted for PA12 coatings with a layer thickness smaller than 250 µm. The characteristic cross-cutting values of 0 to 5 were determined. A value of 0 indicates an excellent bonding between coating and substrate. By contrast, a value of 5 represents a complete delamination of the coating material.

## 3. Results and Discussion

### 3.1. Infrared Thermography

Thermograms, which were made at the end of the deposition process with dependence on the energy density with a powder mass flow of 0.6 g/min, are depicted exemplarily in Figure 3a–c. On the basis of the melting peak temperature (Figure 4a) of the PA12 powder, the areas with temperatures above 186 °C in the thermograms were defined as melt pools. Due to the temporal offset, the deposited material of the previous scan lines is already cooled down to a lower temperature level (<90 °C). For energy densities of 2.1 J/mm^2^ (Figure 3a) and 2.9 J/mm^2^ (Figure 3b), no temperature anomalies on the surface of the deposited PA12 coatings are observed. In contrast to these energy densities, inhomogeneities in the thermogram can be seen for an energy density of 3.6 J/mm^2^ (Figure 3c). These inhomogeneities indicate process instabilities in the PA12 coating due to an excessive energy input.

In general, qualitatively larger melt pools (Figure 3a–c) and higher maximum temperatures (Figure 3d) were determined with an increase in energy density. An increase of the powder mass flow results in lower maximum temperatures, which can be associated with changes of the thermal mass.

The bonding of dissimilar classes of materials is a decisive influencing factor for the process and consolidation characteristics in DED-LB/P. It is well-known from literature [26] that the surface of a metallic substrate is locally fused by the laser beam in DED-LB/M. The metallic powder is simultaneously blown into the melt pool by a powder nozzle. Compared to DED-LB/M, the metallic substrate was only heated by the laser beam in this study due to a lower processing temperature. Besides the direct energy input into the polymer powder by the laser beam, a heat transfer from the heated substrate to the powder material is assumed.

### 3.2. DSC Analysis

Figure 4a compares exemplarily the heating curves of the raw PA12 powder with the PA12 coatings, which were applied by DED-LB/P with a constant powder mass flow of 0.6 g/min for energy densities of 2.1 J/mm^2^, 2.9 J/mm^2^ and 3.6 J/mm^2^. The PA12 powder depicts a sharp and narrow single endothermic melting peak at 186.2 ± 0.1 °C. An endothermic peak at 179.7 ± 1.0 °C with a shoulder at a temperature of approximately 186 °C was detected for coatings processed with an energy density of 2.1 J/mm^2^. The presence of a shoulder on the right is a common phenomenon in PBF-LB/P, which is attributed to unmolten particle cores that have a higher melting temperature than the surrounding spherulite crystalline structures [27]. The reason for the incomplete melting of the PA12 powder in this study is an insufficient heat energy received during DED-LB/P processing.

The coating layers deposited with an increased energy density of 2.9 J/mm^2^ and 3.6 J/mm^2^ exhibit a single endothermic melting peak at 179.8 ± 0.5 °C and 180.0 ± 0.1 °C, respectively. According to Ref. [28], this melting behavior indicates a complete melting of the PA12 particles by the thulium-doped fiber laser. As a consequence of the higher degree of particle melt, improved mechanical properties of the PA12 coatings can be expected [29].

Exothermic transition peaks are observed for the analyzed coatings at a temperature of 160.0 ± 0.8 °C, 161.1 ± 0.4 °C and 161.2 ± 0.3 °C for an energy density of 2.1 J/mm^2^, 2.9 J/mm^2^ and 3.6 J/mm^2^, respectively. These exothermic transition peaks are characteristic for PA12 with low degrees of crystallinity [30] undergoing relaxation of stresses caused by the fast cooling during DED-LB/P processing.

The crystallinity of the PA12 powder was calculated to be 50.5 ± 0.4%. Due to the powder preparation via the dissolution–precipitation process in ethanol at elevated temperatures, the metastable monoclinic α-structure with a high crystallinity is formed in the PA12 powder [31]. Compared to the PA12 powder, the deposited PA12 coatings exhibit a lower degree of crystallinity (22.8 ± 1.8% − 30.2 ± 0.9%), which is shown in Figure 4b. Similar to PBF-LB/P [28], the decrease in crystallinity from PA12 powder to the coatings is associated with a phase transformation to the stable γ-structure during DED-LB/P processing. Furthermore, we assume the suppression of crystalline phases due to high cooling rates in DED-LB/P, which results in higher amounts of amorphous regions. Compared to energy densities of 2.9 J/mm^2^ and 3.6 J/mm^2^, higher degrees of crystallinity in the range between 27.0 ± 1.3% and 30.2 ± 0.9% are observed for an energy density of 2.1 J/mm^2^. When the energy density is increased to 2.9 J/mm^2^ and 3.6 J/mm^2^ for coatings deposited with a powder mass flow of 0.6 g/min, the crystallinity is reduced by approximately 12% and 13%, respectively. The lower crystallinity for energy densities of 2.9 J/mm^2^ and 3.6 J/mm^2^ is attributed to a higher degree of particle melt. There is no evidence for the change in crystallinity with respect to the powder mass flow because results are in the range of the standard deviation.

### 3.3. Layer Height and Surface Topography

Figure 5 shows the layer thickness of the coatings depending on the energy density for different powder mass flows, which was measured by means of the dial gauge. For energy densities of 2.1 J/mm^2^ and 2.9 J/mm^2^, reproducible, firmly adhering PA12 coating layers were deposited, which fully covered the stainless-steel substrate. Layer thicknesses in the range from 0.14 ± 0.01 to 0.54 ± 0.04 mm can be achieved for these energy densities by adjusting the powder mass flow.

When the energy density was raised from 2.1 J/mm^2^ to 2.9 J/mm^2^, an increase by 86%, 84% and 139% in the layer height is observed for a powder mass flow of 0.2 g/min, 0.6 g/min and 1.4 g/min, respectively. Due to the higher energy input, an increased proportion of the powder particle stream was melted in the laser–powder interaction zone (Figure 2b), which is detectable in larger melt pools in the thermograms (Figure 3a–c). In this regard, the adjustment of the powder mass flow shows a higher influence on the layer thickness for an energy density of 2.9 J/mm^2^ compared to 2.1 J/mm^2^, whereas, for an energy density of 2.1 J/mm^2^, the layer thickness increased by 58% due to a raise of the powder mass flow from 0.2 g/min to 1.4 g/min, the layer height increased by 103% for an energy density of 2.9 J/mm^2^.

The relative standard deviations of the measured layer thicknesses for the coatings consolidated by energy densities of 2.1 J/mm^2^ and 2.9 J/mm^2^ were in the range from 5.7% to 8.6%. By contrast, the coatings deposited by an energy density of 3.6 J/mm^2^ show higher relative standard deviations with values up to 37.4%, which correlate to the observed inhomogeneities in the thermogram (Figure 3c).

Photographic images and corresponding 3D color-coded height maps of the PA12 coating surfaces with dependence on the energy density are shown exemplarily for a powder mass flow of 0.6 g/min in Figure 6. Surface irregularities are identified in the 3D height map for an energy density of 2.1 J/mm^2^ (Figure 6a), which are attributed to unmolten powder particles. The presence of a shoulder peak at higher temperatures during DSC heating experiments (Figure 4a) confirms this observation. Poor surface quality with a high roughness is a well-known problem in DED-LB/M [32]. For improving the surface finish, post-processes such as milling or laser remelting are necessary, which adversely increase production time and costs [33].

When increasing the energy density from 2.1 J/mm^2^ to 2.9 J/mm^2^, a smoother surface with a smaller proportion of unmolten powder particles is observed in the 3D height map (Figure 6b). In accordance with the measurements of the layer thickness, a laser exposure with an energy density of 3.6 J/mm^2^ results in PA12 coatings exhibiting a heavily rugged surface topography (Figure 6c). An excessive energy input and the related increase of the temperature of the melt pool are identified as reasons for instabilities during DED-LB/P processing. According to the Arrhenius-type equation [34], the viscosity decreases exponentially with increasing temperature. Due to the lower viscosity at high temperatures, it is assumed that the mixed stream of polymer powder and carrier gas (argon) formed by the nozzle leads to turbulences in the melt pool. As a consequence of the turbulences, the rugged surface topography of the coating is unavoidable for higher energy densities when processing PA12. A compromise must be found between the degree of particle melt and process stability.

Since the rugged surface can be recognized by inhomogeneities in the thermogram (Figure 3c), the use of an IR camera represents a viable approach for in-situ monitoring in DED-LB/P. The thermal contrast in the thermograms can be associated with a local variation of the emissivity. According to literature [35], the surface roughness has a decisive influence on the emissivity. Internal reflections within the valleys of the surface are regarded as a reason for an increase of the emissivity [36].

### 3.4. Porosity of the PA12 Coatings

The quantitative analysis of the porosity of the cross-sections, as seen in Figure 7, presents a clear differentiability among the energy densities. Due to the heavily rugged surface topography, the coatings consolidated by an energy density of 3.6 J/mm^2^ are not analyzed in the further course of this study. While a porosity of 2.3 ± 0.6% is achievable for coatings deposited with an energy density of 2.1 J/mm^2^ and a powder mass flow of 0.6 g/min, results show a porosity up to 12.1 ± 6.4% for an energy density of 2.9 J/mm^2^ and a powder mass flow of 0.2 g/min. The examined powder mass flows (0.2–1.4 g/min) show no significant influence on the porosity. Microscopic images of polished cross-sections are shown exemplarily in Figure 8 for energy densities of 2.1 J/mm^2^ and 2.9 J/mm^2^. For both energy densities, the porosity is higher at the lower half of the coating. In the coating consolidated by an energy density of 2.9 J/mm^2^, tubular pores are formed, which are substantially larger than the spherical pores.

Due to the high complexity of the DED-LB process, the underlying causes behind the increased porosity for an energy density of 2.9 J/mm^2^ can be manifold. Since an energy density of 2.9 J/mm^2^ leads to higher layer thicknesses compared to an energy density of 2.1 J/mm^2^ (Figure 5), the optical penetration depth might be the limiting factor. Therefore, pores near the coating–substrate interface cannot be completely eliminated in coatings with a higher layer thickness. Analogously to the turbulences in the melt pool, the temperature dependency of the melt viscosity of the polymer plays another considerable role with regard to the higher porosity for an energy density of 2.9 J/mm^2^. As a consequence of the lower viscosity at high temperatures, the process gases penetrate more easily into the lower layers of the molten zone. In order to prove this hypothesis and fully understand the consolidation mechanisms, further investigations on the viscosity of the polymer and the carrier gas flow as well as shielding gas flow are required.

### 3.5. Coating Adhesion

Regardless of the energy density and powder mass flow, no delamination at the interface between the PA12 coating layer and the stainless steel substrate is observed (Figure 8). This is an important prerequisite for an adequate adhesion strength of the coating.

According to DIN EN ISO 2409, the maximum layer thickness for cross-cutting tests corresponds to 250 µm. Therefore, PA12 coatings deposited with an energy density of 2.9 J/mm^2^ and 3.6 J/mm^2^ were not examined. The microscopic images of PA12 coatings after the cross-cutting tests in Figure 9 demonstrate that none of the squares of the lattice is detached. Therefore, the examined samples deposited with powder mass flows between 0.2 g/min and 1.4 g/min exhibit cross-cutting values of 0, which indicate a good adhesion between the PA12 coating and stainless-steel substrate.

## 4. Conclusions

In the present study, PA12 coatings on stainless steel substrates were generated by DED-LB/P. Compared to conventional coating processes (e.g., electrostatic powder spraying and flame spraying), a shortening of the process chain and a spatially resolved polymer deposition can be achieved. Due to characteristic absorption bands, a thulium-doped fiber laser with a wavelength of 1.94 µm was used in a DED-LB/P setup to consolidate PA12 coatings. In the following, the key findings are summarized:As a higher number, powder particles are melted in the laser–powder interaction zone, and an increase of the layer height is reached by raising the energy density and powder mass flow. Layer thicknesses in the range from 0.14 ± 0.01 mm to 0.54 ± 0.04 mm can be reproducibly adjusted;PA12 coatings exhibiting a porosity of 2.3 ± 0.6% can be achieved by DED-LB/P with an energy density of 2.1 J/mm^2^. Cross-cutting tests indicate a good bonding between the coating layers and the stainless-steel substrate. The PA12 coatings consolidated by an energy density of 2.1 J/mm^2^ exhibit a poor surface quality due to unmolten powder particles;Smoother surfaces of the PA12 coating are achievable by increasing the energy density to 2.9 J/mm^2^, leading to a higher degree of particle melt. However, the improvement of the surface quality comes with an increase in porosity (up to 12.1 ± 6.4%);An excessive energy input ($E_{D}$ = 3.6 J/mm^2^) results in process instabilities, which cause a heavily rugged surface topography. Infrared thermography is suitable for in-situ detection of these process defects in DED-LB/P.

Based on these results, the feasibility of an absorber-free DED-LB/P process by using a thulium-doped fiber laser at a wavelength of 1.94 µm is proven. This enables new fields of applications, particularly in the medical sector. The focus of future works will comprise the improvement of the surface quality by a remelting procedure with the laser beam. In addition, other polymer powders (e.g., PEEK) and the manufacturing of multilayer structures will be investigated.

## Figures and Tables

**Figure 1 polymers-14-03729-f001:**
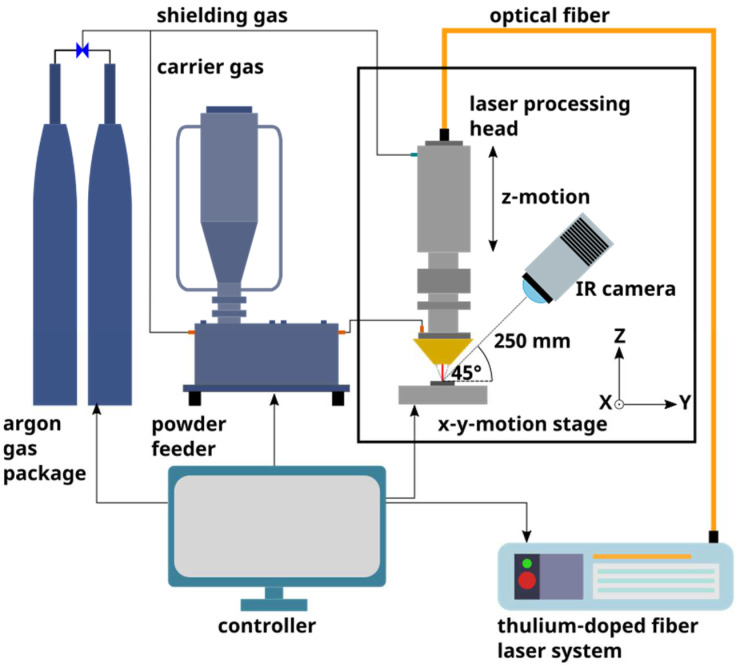
Schematic of the DED-LB/P setup for generating polymer coatings on stainless steel substrates.

**Figure 2 polymers-14-03729-f002:**
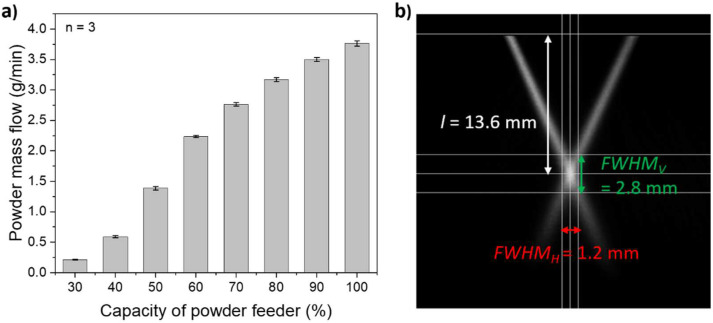
(**a**) Experimentally determined powder mass flow of PA12 powder depending on the capacity of the powder feeder (*n* = 3); (**b**) high-speed camera analysis of the powder stream for a carrier gas flow of 6 L/min, shielding gas flow of 10 L/min and a powder mass flow of 0.6 g/min.

**Figure 3 polymers-14-03729-f003:**
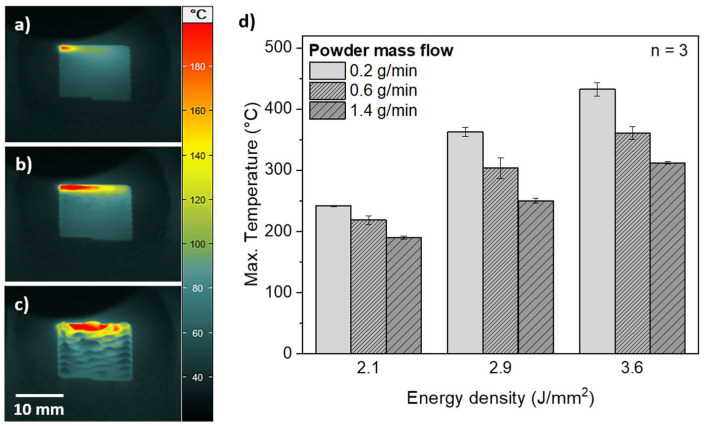
Color-coded thermographic mappings of PA12 coatings deposited by a powder mass flow of 0.6 g/min for energy densities of (**a**) 2.1 J/mm^2^, (**b**) 2.9 J/mm^2^ and (**c**) 3.6 J/mm^2^. The maximum temperatures during DED-LB/P processing as a function of the energy density and powder mass flow (*n* = 3) are shown in (**d**) (emission coefficient set to 0.98).

**Figure 4 polymers-14-03729-f004:**
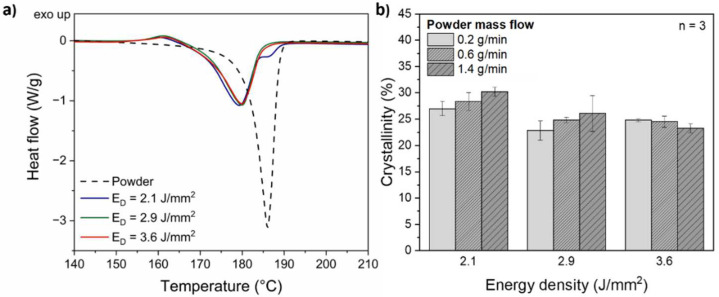
(**a**) Representative DSC heating curves of PA12 powder and coatings applied by DED-LB/P with a constant powder mass flow of 0.6 g/min for energy densities of 2.1 J/mm^2^, 2.9 J/mm^2^ and 3.6 J/mm^2^. The heating rate was 10 K/min; (**b**) crystallinity of the PA12 coatings as a function of the energy density and powder mass flow (*n* = 3).

**Figure 5 polymers-14-03729-f005:**
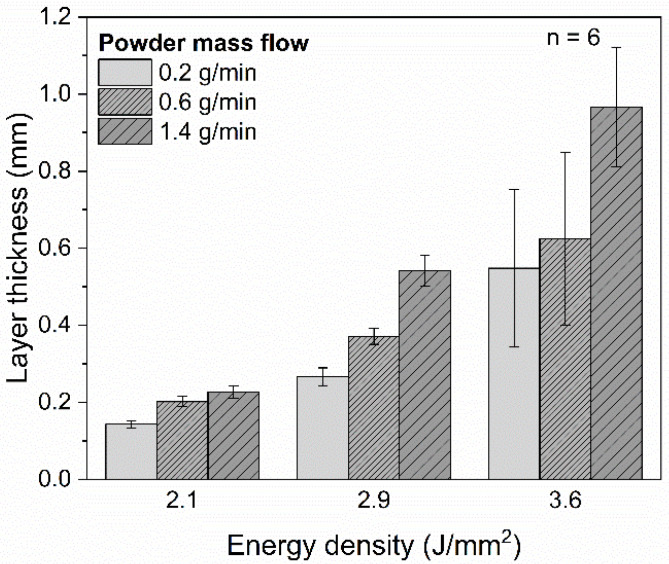
Layer thickness of the PA12 coatings on stainless steel substrates as a function of the energy density for powder mass flows of 0.2 g/min, 0.6 g/min and 1.4 g/min (*n* = 6).

**Figure 6 polymers-14-03729-f006:**
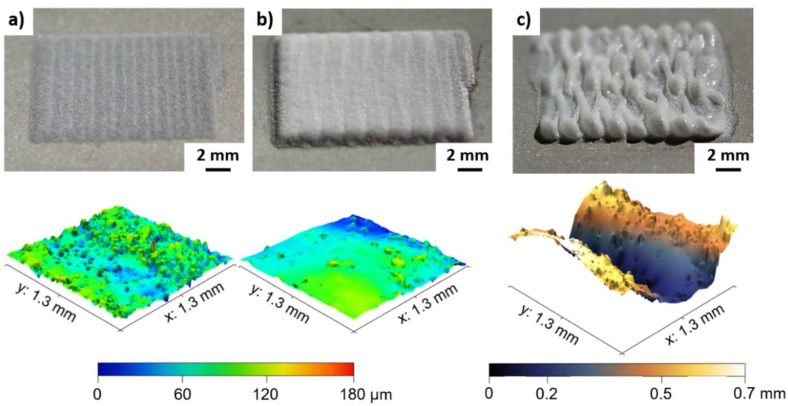
Photographic images (top) and corresponding 3D view of color-coded height maps (bottom) show surface topography of PA12 coatings deposited by a powder mass flow of 0.6 g/min for energy densities of (**a**) 2.1 J/mm^2^, (**b**) 2.9 J/mm^2^ and (**c**) 3.6 J/mm^2^.

**Figure 7 polymers-14-03729-f007:**
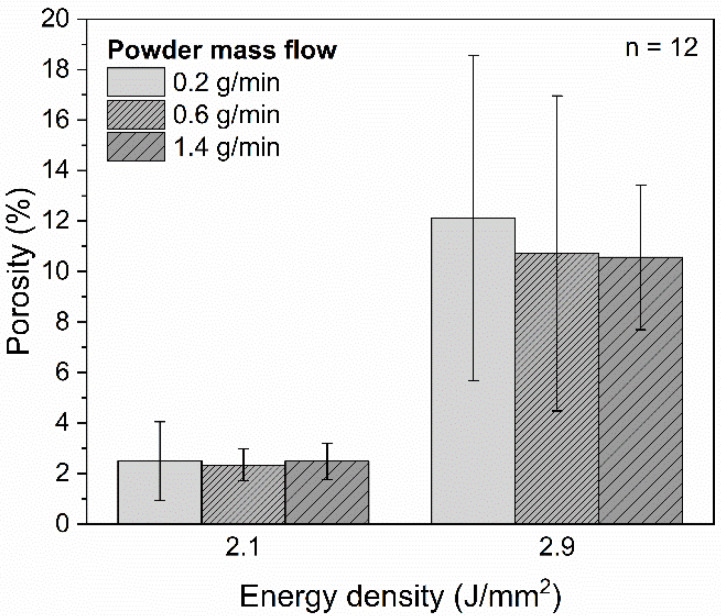
Porosity of the PA12 coatings as a function of on the energy density for powder mass flows of 0.2 g/min, 0.6 g/min and 1.4 g/min (*n* = 12).

**Figure 8 polymers-14-03729-f008:**
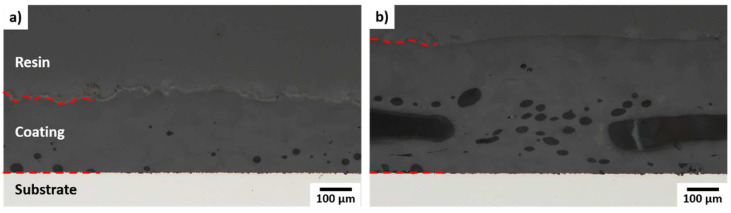
Exemplary microscopic images of polished cross-sections of PA12 coatings deposited by a powder mass flow of 0.6 g/min and different energy densities of (**a**) 2.1 J/mm^2^ and (**b**) 2.9 J/mm^2^.

**Figure 9 polymers-14-03729-f009:**
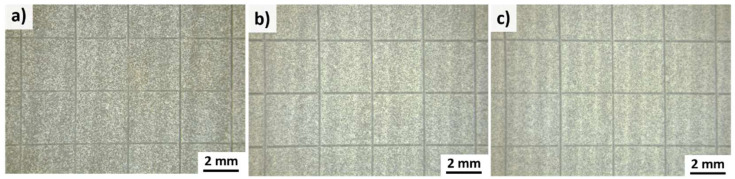
Microscopic images after the cross-cutting tests. PA12 coatings deposited by a constant energy density of 2.1 J/mm^2^ and different powder mass flows of (**a**) 0.2 g/min, (**b**) 0.6 g/min and (**c**) 1.4 g/min.

**Table 1 polymers-14-03729-t001:** Experimental design for the generation of PA12 coatings by DED-LB/P.

Parameter	Unit	Value
Hatch distance h	mm	0.9
Feed rate v	mm/s	5.0
Beam diameter d	mm	1.9
$\mathrm{Carrier}\;\mathrm{gas}\;\mathrm{flow}\;V_{C}$	L/min	6.0
$\mathrm{Shielding}\;\mathrm{gas}\;\mathrm{flow}\;V_{S}$	L/min	10.0
Working distance l	mm	13.6
Laser power P	W	6.0/13.1/20.2/27.3/34.4
$\mathrm{Energy}\;\mathrm{density}\;E_{D}$	J/mm^2^	0.6/1.4/2.1/2.9/3.6
Powder mass flow dm/dt	g/min	0.2/0.6/1.4

## Data Availability

The data presented in this study are available on request from the corresponding author.

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
