# Peer review of "Generation of Polyamide 12 Coatings on Stainless Steel Substrates by Directed Energy Deposition with a Thulium-Doped Fiber Laser (DED-LB/P)"

_polymers, 2022, doi:10.3390/polym14183729_

Round 1

Reviewer 1 Report

The submitted paper deals with the use of a thulium-doped fiber laser with a wavelength of 1.94 μm in a DED-LB/P setup to consolidate PA12 coatings on stainless steel substrates. It’s an interesting and innovative work. The paper is well-written and the conclusions are well-supported. My main remarks are the followings:

1.              The authors mentioned in the manuscript that the substrate faces were sandblasted to improve the adhesion between the substrate and coating. Please mention the applied conditions during sandblasting. Moreover, the reasons which lead to an improved film adhesion have to be mentioned. In this context, some details verifying it have to be presented i.e. roughness increase indicating a better coating-substrate interlocking.

2.              Have the authors conducted some mechanical tests for evaluating the film adhesion i.e. scratch tests or Rockwell C indentations etc? Although no delamination at the interface between the PA12 coating layer and the stainless steel substrate was observed and this is a good indication for film adhesion, the response of the coating-substrate interface under loading is necessary for assessing the quality of the film adhesion.

3.              Since these coatings appear a porosity, have the authors evaluated the film creep behavior for various deposition conditions by conducting a mechanical test? The Knowledge of the creep behavior is important when such coatings are going to be applied in real applications.

Reviewer 2 Report

In this paper, the preparation, detection and data discussion of PA12 coating are very perfect, which can well reflect the advanced nature and preparation controllability of the coating, and fully meet the requirements of the journal publication.

1. Remove the black border on the left side of Figure 1.

2. Letters in the formula are identified in italics.

Round 2

Reviewer 1 Report

The authors made all the appropriate changes and the paper can be published in the current form.